# Identification of Genes Whose Expression Overlaps Age Boundaries and Correlates with Risk Groups in Paediatric and Adult Acute Myeloid Leukaemia

**DOI:** 10.3390/cancers12102769

**Published:** 2020-09-27

**Authors:** Lindsay Davis, Ken I. Mills, Kim H. Orchard, Barbara-Ann Guinn

**Affiliations:** 1Patrick G. Johnson Centre for Cancer Research, Queen’s University Belfast, Lisburn Road, Belfast BT9 7AE, UK; ldavis05@qub.ac.uk (L.D.); k.mills@qub.ac.uk (K.I.M.); 2Department of Haematology, University Hospital Southampton NHS Foundation Trust, Tremona Road, Southampton SO16 6YD, UK; kho@soton.ac.uk; 3Department of Biomedical Sciences, University of Hull, Cottingham Road, Kingston-upon-Hull HU6 7RX, UK

**Keywords:** paediatric and adult acute myeloid leukaemia, antigen identification, meta-analysis, mRNA-sequencing, *WNT*, *HOX*, *SOX*

## Abstract

**Simple Summary:**

To better understand whether acute myeloid leukaemia differs between children and adults, we have analysed the expression of genes in samples from both patient groups. Using previously published data, we compared gene expression between patient risk subgroups. We examined patients who had a poor chance of survival, based on clinical assessments, and those with a good chance of survival, to see whether there was any difference in the genes expressed in their leukaemic cells. Then we compared the genes on these lists between adults and children with acute myeloid leukaemia. We believe that patients with good or poor survival chances express genes that provide insights into how leukaemic cells behave. We hope that this work will provide new information about the mechanisms that underlie acute myeloid leukaemia and answer questions on the ways this form of leukaemia is similar in adults and children, which will then tell us whether the same treatments could be used for both age groups of patients.

**Abstract:**

Few studies have compared gene expression in paediatric and adult acute myeloid leukaemia (AML). In this study, we have analysed mRNA-sequencing data from two publicly accessible databases: (1) National Cancer Institute’s Therapeutically Applicable Research to Generate Effective Treatments (NCI-TARGET), examining paediatric patients, and (2) The Cancer Genome Atlas (TCGA), examining adult patients with AML. With a particular focus on 144 known tumour antigens, we identified *STEAP1*, *SAGE1*, *MORC4*, *SLC34A2* and *CEACAM3* as significantly different in their expression between standard and low risk paediatric AML patient subgroups, as well as between poor and good, and intermediate and good risk adult AML patient subgroups. We found significant differences in event-free survival (EFS) in paediatric AML patients, when comparing standard and low risk subgroups, and quartile expression levels of *BIRC5*, *MAGEF1*, *MELTF*, *STEAP1* and *VGLL4*. We found significant differences in EFS in adult AML patients when comparing intermediate and good, and poor and good risk adult AML patient subgroups and quartile expression levels of *MORC4* and *SAGE1*, respectively. When examining Kyoto Encyclopedia of Genes and Genomes (KEGG) (2016) pathway data, we found that genes altered in AML were involved in key processes such as the evasion of apoptosis (*BIRC5*, *WNT1*) or the control of cell proliferation (*SSX2IP*, *AML1-ETO*). For the first time we have compared gene expression in paediatric AML patients with that of adult AML patients. This study provides unique insights into the differences and similarities in the gene expression that underlies AML, the genes that are significantly differently expressed between risk subgroups, and provides new insights into the molecular pathways involved in AML pathogenesis.

## 1. Introduction

Acute myeloid leukaemia (AML) is the second most frequent haematological malignancy in the paediatric population and remains a leading cause of childhood cancer mortality. With little improvement in the last few decades, survival remains around 70% [1] despite treatments that include maximally intensive chemotherapies and myeloablative haematopoietic stem cell transplantation. The anti-leukaemia effect mediated by the lymphocytes and natural killer (NK) cells of the donor immune system has been established in haematopoietic stem cell transplantation, and also as adoptive immunotherapy after consolidation chemotherapy schemes, but the most common cause of death remains relapse. 

Part of the difficulty in treating AML is caused by its clinical and molecular heterogeneity [2] with a range of cytogenetic rearrangements having been described (reviewed recently by [3]). There is a clear association between age, genetic mutations, cytogenetic rearrangements and survival [1], but few genetic abnormalities appear in more than 20% of patients [3]. Diagnostic and prognostic markers of AML such as *KIT* [4,5], *FLT3* [6], *CEBPA* [7,8] and *NPM1* [9] have been identified in both adult and paediatric AML, although few of the tumour antigens known in adult AML (recently reviewed in [10]) have been investigated in paediatric AML (reviewed in [11]). The development of strategies that target AML cells has been hindered by the lack of an antigen with high specificity for blasts that is not present on normal haematopoietic stem cells, and thus would not result in myelotoxicity. In the absence of this, targeting antigens that are expressed on or by AML cells, but not essential tissues would be a tolerable, if less favourable, alternative. 

Since 2017, there has been a rapid expansion in new therapies available to paediatric AML patients for whom chemotherapy has been ineffective or intolerable (reviewed recently in [3,12]). Most have focused on antibody–drug conjugates, bi-specific T cell engagers and chimeric antigen receptor (CAR)-T cells (reviewed in [13]). Antibody-based targeting of leukaemia-associated antigens (LAAs) offers the opportunity to deplete target malignant blasts due to their exquisite specificity for the target surface antigen, creating an ‘on-target/on-tumour effect’. The disadvantage arises due to depletion of antigen-expressing healthy cells that also express the antigen: the so-called ‘on-target/off-tumour effect’. More recent developments of CAR-T cells and CAR-NK cells have been transformative, providing cures to children with poor survival prospects, but patients can suffer life-threatening toxicities, including neurologic dysfunction, cytokine release syndrome and macrophage activation syndrome. These side effects are due to off-target/off-tumour effects related to the rapid activation of CAR-T cells soon after infusion, leading to a release of inflammatory cytokines, which a change in antigen target would not necessarily circumvent. Perna et al. [14] demonstrated the use of transcriptomics and proteomics to identify pairs of antigens uniquely expressed on AML blasts [14], these CAR-T cell co-targets were proposed as one way to circumvent the toxicity to non-haematopoietic tissues in the future. 

Despite the improvements to patient outcomes and associated overall survival (OS) rates they offer, current therapies still have limitations including the augmented immune response enabled by immune facilitators and checkpoint inhibitors such as those targeting cytotoxic T-lymphocyte associated protein 4 (CTLA-4) and programmed cell death protein-1 (PD-1), respectively, leading to a unique group of side effects called immune-related adverse events (reviewed in [15]). It is essential, therefore, that we identify new targets for therapy so that we can widen the scope of future treatments and determine their relevance to paediatric and adult leukaemia. In this study, we have analysed two publicly available databases and the mRNA sequencing data therein. The National Cancer Institute’s Therapeutically Applicable Research to Generate Effective Treatments (NCI-TARGET) initiative recruited children and young adults with AML taking part in the Children’s Oncology Group studies to better characterise the molecular genetic markers of the disease. The second database was The Cancer Genome Atlas (TCGA), a collaboration between the National Cancer Institute and the National Human Genome Research Institute, aimed at improving the characterisation of multiple cancers, including 200 adult patients with AML in the TCGA-LAML project. Comprehensive clinical data were collected as part of the development of both databases and included information on gender, age at diagnosis, race, haematological indices, treatments received, and prognostic events, as well as selected cytogenetic abnormalities and other molecular features. 

## 2. Results

### 2.1. Differential Gene Expression (DGE) Analysis

Good versus poor prognostic subgroup pairs were formed from the TARGET AML data and then from the TCGA AML data (Figure 1A) so that they could be compared by differential gene expression (DGE) analysis (Figure 1B). Event-free survival (EFS) probability distributions were found to be significantly different (log-rank; *p* < 0.05) between the TARGET (Figure 2A) standard and low risk subgroups (*p* = 0.02), and also significantly different between the TCGA (Figure 2B) intermediate and good risk subgroups, and poor and good risk subgroups (*p* = 0.01 and *p* = 0.007, respectively). The EFS of TCGA patients < 60 years of age and ≥60 years of age within total TCGA patients, within intermediate and good risk subgroups, and within poor and good risk subgroups, were also compared in pairwise fashion (Appendix A). EFS probability distributions were found to be significantly different between TCGA patients < 60 years of age and ≥60 years of age within total TCGA patients; however, no significant results were observed between risk subgroups for patients <60 years of age and ≥60 years of age. 

This analysis resulted in the identification of significant differentially expressed genes (DEGs) (Benjamini–Hochberg (BH)-adjusted *p* value < 0.01) for each comparison and allowed for the identification of significant DEGs that also appeared on a list of genes of interest (GOI) which was formed prior to analysis (Figure 1B and Figure 2C,D).

#### 2.1.1. TARGET AML Data

From the paediatric TARGET dataset, eight prognostic subgroup pairs were formed for DGE analysis and compared in a pairwise fashion (Appendix A); however, three comparisons did not show any GOI to be significantly differentially expressed: EFS < 1 year versus EFS > 5 years within the total patient group, OS < 1 year versus OS > 5 years within the total patient group, and EFS < 6 months versus EFS > 2 years within the patients possessing mixed-lineage leukaemia (*MLL*, also known as *KMT2A*) rearrangements as their primary cytogenetic abnormality. Four comparisons of the remaining five had subgroups containing < 10 patients; therefore, analysis was continued but without a focus on their results. The standard versus low risk comparison did, however, identify significantly differentially expressed GOI and consisted of sufficient patient numbers.

Antigens on the GOI list that were significantly differentially expressed between the standard and low risk patient subgroups (*n* = 31 vs. *n* = 31) (Table 1) included carcinoembryonic antigen-related cell adhesion molecule 3 (*CEACAM3*; also known as CD66d), *CEACAM6* (also known as CD66c), *CEACAM8* (also known as CD66b), Folate Hydrolase 1 (*FOLH1*), MORC Family CW-Type Zinc Finger 4 (*MORC4*), sarcoma antigen 1 (*SAGE1*; *CT14*), Solute Carrier Family 34 Member 2 (*SLC34A2*) and the metalloreductase six-transmembrane epithelial antigen of prostate member 1 (*STEAP1*). 

Of the other notable genes not present on the original list of 144 GOI, we found significant differences in the expression of numerous sex-determining region on the Y chromosome (SRY)-related high mobility group (HMG)-box (*SOX*) genes between patient subgroups. Most comparisons had to be excluded due to low patient numbers (<10) in at least one subgroup. However, within the standard versus low risk subgroup comparison that was not excluded due to insufficient patient numbers, *SOX15* (*p* = 7.94 × 10^−9^), *SOX11* (*p* = 2.45 × 10^−8^), *SOX5* (*p* = 6.16 × 10^−5^), *SOX30* (*p* = 3.86 × 10^−4^), *SOX18* (*p* = 4.93 × 10^−4^) and *SOX8* (*p* = 3.69 × 10^−3^) expression was significantly different. The homeobox (HOX) genes’ transcription differed significantly between standard and low risk subgroups (*HOXA-AS3 p* = 3.79 × 10^−19^; *HOXB-AS3 p* = 1.01 × 10^−10^; *HOXA10-AS p* = 3.65 × 10^−4^; *HOXB7 p* = 1.13 × 10^−10^; *HOXA10 p* = 1.15 × 10^−7^; *HOXA7 p* = 2.05 × 10^−7^), as did the cluster of differentiation antigens, CD34, CD52 and CD99, with adjusted *p* values of 1.53 × 10^−7^, 2.79 × 10^−12^ and 2.48 × 10^−8^, respectively.

The transcription of fourteen tripartite motif family (TRIM) genes, including *TRIM47* (*p* = 3.25 × 10^−9^), *TRIM71* (*p* = 4.81 × 10^−7^), *TRIM9* (*p* = 2.82 × 10^−6^), *TRIM29* (*p* = 3.33 × 10^−6^), *TRIM7* (*p* = 3.84 × 10^−6^), and *TRIM24* (*p* = 2.33 × 10^−5^), was also significantly different between standard and low risk subgroups.

#### 2.1.2. TCGA AML Data

Five prognostic subgroup pairs were compared by DGE analysis from the TCGA adult AML data (Appendix A); however, three comparisons did not show any GOI to be significantly differentially expressed and had subgroups with insufficient patient numbers (<10): EFS < 1 year versus EFS > 5 years within the total patient group, OS < 1 year versus OS > 5 years within the total patient group, and EFS < 6 months versus EFS > 2 years within the patients with normal primary cytogenetics. The intermediate versus good risk and poor versus good risk comparisons did, however, identify significantly differentially expressed GOI and consisted of sufficient patient numbers.

The significant DEGs in the remaining two comparisons included GOI such as the coiled coil domain, containing 186 (*CCDC186*), *CEACAM3*, *CEACAM6*, *CEACAM8*, *MORC4*, Myosin heavy chain 11 (*MYH11*), *SAGE1*, *SLC34A2* and *STEAP1* in the poor versus good risk subgroup comparison (*n* = 32 vs. *n* = 17) and *CCDC186*, *CEACAM3*, *FOLH1*, *MORC4*, *MYH11*, *SAGE1*, *SLC34A2* and *STEAP1* in the intermediate versus good risk subgroup comparison (*n* = 75 vs. *n* = 17) (Table 1).

Of the other notable genes not present on the original list of 144 GOI, we found 19 HOX genes that were significantly differentially expressed between poor and good, and intermediate and good risk subgroups, including *HOXA6* (*p* = 7.87 × 10^−32^; *p* = 1.04 × 10^−34^), *HOXA3* (*p* = 3.57 × 10^−26^; *p* = 6.50 × 10^−33^) and *HOXA7* (*p* = 5.05 × 10^−16^; *p* = 1.19 × 10^−24^), respectively. In addition, there were five HOX antisense (AS) genes whose expression differed significantly between poor and good, and intermediate and good risk subgroups. These were *HOXA-AS3* (*p* = 3.19 × 10^−33^; *p* = 1.30 × 10^−39^), *HOXA-AS2* (*p* = 9.73 × 10^−26^; *p* = 6.77 × 10^−35^), *HOXB-AS3* (*p* = 1.02 × 10^−5^; *p* = 3.85 × 1^−27^), *HOXA10-AS* (*p* = 7.04 × 10^−11^; *p* = 8.13 × 10^−13^) and *HOXA11-AS* (*p* = 1.52 × 10^−5^; *p* = 1.96 × 10^−11^), respectively.

Runt-related transcription factor-1 (*RUNX1*), Partner Transcriptional Co-Repressor 1 (*RUNX1T1*) (*p* = 2.92 × 10^−17^), *RUNX3* (*p* = 6.50 × 10^−12^) and the Histamine N-Methyltransferase (*HNMT*) (*p* = 5.48 × 10^−10^) genes were each differentially expressed between intermediate and good risk subgroup patients. The transcription of ten TRIM genes, including *TRIM9* (*p* = 7.94 × 10^−13^), *TRIM47* (*p* = 7.03 × 10^−10^), *TRIM29* (*p* = 1.20 × 10^−7^), *TRIM10* (*p* = 2.77 × 10^−7^), *TRIM71* (*p* = 3.94 × 10^−5^), *TRIM8* (*p* = 8.27 × 10^−5^), *TRIM15* (*p* = 2.87 × 10^−4^), *TRIM6* (*p* = 5.88 × 10^−4^), *TRIM24* (*p* = 3.49 × 10^−3^), and *TRIM68* (*p* = 5.52 × 10^−3^), were significantly different between intermediate and good risk subgroups, as were the transcription of *PRICKLE1* and *PRICKLE2* (*p* = 5.06 × 10^−9^ and *p* = 1.56 × 10^−5^, respectively), both of which play roles in the WNT signalling pathway. *RUNX3* (*p* = 1.67 × 10^−14^), *WNT6* (*p* = 3.99 × 10^−10^) and eight members of the TRIM family, including *TRIM9* (*p* = 2.00 × 10^−9^), *TRIM10* (*p* = 4.65 × 10^−8^), *TRIM24* (*p* = 1.34 × 10^−6^) and *TRIM29* (*p* = 6.26 × 10^−6^), were significantly differentially expressed between the poor and good risk subgroups.

Members of the SOX family also appeared as significant DEGs with the TCGA data. In the poor versus good risk comparison, *SOX18* (*p* = 2.01 × 10^−4^), *SOX5* (*p* = 1.65 × 10^−3^) and *SOX15* (*p* = 7.94 × 10^−3^) were found to be significant DEGs, while *SOX6* was found to be a significant DEG between both intermediate versus good, and poor versus good risk comparisons (*p* = 6.86 × 10^−4^ and *p* = 1.61 × 10^−5^, respectively). Of the cluster of differentiation antigens, *CD59* and *CD82* were found to be significantly differentially expressed when comparing poor and good risk subgroups in adult AML (*p* = 2.09 × 10^−12^ and *p* = 6.60 × 10^−11^, respectively). In addition, *CD19* (*p* = 2.52 × 10^−6^), *CD7* (*p* = 7.27 × 10^−6^), *CD81* (*p* = 1.67 × 10^−5^), *CD276* (*p* = 1.67 × 10^−4^), and *CD70* (*p* = 1.80 × 10^−4^), were each significantly differentially expressed between poor and good risk subgroups in adult AML. 

High mobility group (HMG) box family member 2 (*TOX2*) (*p* = 5.61 × 10^−6^), and other members of the TOX subfamily of transcription factors, are almost identical to HMG-box DNA-binding domains which function to modify the chromatin structure; interleukin-7 (*IL7*) (*p* = 3.14 × 10^−21^), a pro-inflammatory cytokine, *IL10* (*p* = 6.55 × 10^−7^), levels of which have previously been shown to directly correlate with survival in adult AML [16], and *IL15* (*p* = 1.21 × 10^−5^), shown to enhance NK cytotoxicity in patients with AML by upregulating activating NK cell receptors [17], each showed significantly different transcription when comparing the poor and good risk subgroups.

### 2.2. Survival Analysis

Where a subgroup comparison identified a significantly differentially expressed GOI, total raw counts were transformed across all patients in the comparison and quartile thresholds for each GOI were calculated. This allowed for the transformed expression value of each gene in a patient sample to be assigned to a quartile (Q1/Q2/Q3/Q4). Log-rank tests were used to assess whether there were differences in EFS between patients with expression levels in Q1 and patients with expression levels in Q4 for each significantly differentially expressed GOI. Log-rank tests were also used to assess differences in EFS between patients with expression levels in Q2 and expression levels in Q3 (Figure 3).

#### 2.2.1. TARGET AML Data

Log-rank test results were significant (*p* value < 0.05) for five significantly differentially expressed GOI (Table 2). Significant differences in the EFS of patients were found when comparing standard and low risk subgroups, and Q1 and Q4 expression levels of the protein coding gene Baculoviral Inhibitor Apoptosis Proteins (IAP) Repeat Containing 5 (*BIRC5*) (*p* = 0.03; Figure 3A(i)), melanoma antigen family F1 (*MAGEF1*) (*p* = 0.04; Figure 3A(ii)), Melanotransferrin (*MELTF*) (*p* = 0.01; Figure 3A(iii)) and *STEAP1* (*p* = 0.03; Figure 3A(iv)) and when comparing Q2 and Q3 expression levels of Vestigial-like 4 (*VGLL4*) across standard and low risk subgroups (*p* = 0.04; Figure 3A(v)). Higher EFS probabilities were observed over time in patients who had Q1 expression levels of *BIRC5*, *MAGEF1*, and *STEAP1* than with patients who had Q4 expression levels, whereas, with *MELTF*, higher EFS probabilities were observed over time with patients who had Q4 expression levels than with patients who had Q1 expression levels. With *VGLL4*, higher EFS probabilities were observed over time with patients who had Q2 expression levels than with patients who had Q3 expression levels.

#### 2.2.2. TCGA AML Data

Significant differences in the EFS of patients were found with two significantly differentially expressed GOI: *MORC4* across patients in intermediate and good risk subgroups (*p* = 0.02; Figure 3B(i)) and *SAGE1* across patients in poor and good subgroups (*p* = 0.04; Figure 3B(ii)), when comparing Q1 and Q4 expression levels (log-rank test; *p* value < 0.05) (Table 2). Higher EFS probabilities were observed over time with patients across the intermediate and good risk subgroups who had Q4 expression levels of *MORC4* than with patients who had Q1 expression levels. Higher EFS probabilities were observed over time with patients across the poor and good risk subgroups who had Q1 expression levels of *SAGE1* than with patients who had Q4 expression levels.

### 2.3. Pathway Analysis

The most common KEGG (2016) terms overall in both the TARGET and TCGA data, using the total (Table 3A), downregulated only (log2 FC < 0) (Table 3B), and upregulated only (log2 FC > 0), significantly differentially expressed GOI (Table 3C) were identified. Appendix A shows all overlapping genes for each term along with the number of prognostic subgroup comparisons where the term appeared within the top ten. The results were ordered by Enrichr’s combined score and the top ten KEGG terms were noted (where there were more than ten). In the TARGET comparison (standard versus low risk) for all significantly differentially expressed GOI, Fanconi’s anaemia pathway (*FANCC*, *BRCA2*) and pathways in cancer (*DAPK1*, *BIRC5* and *BRCA2*) highlighted genes already known to play key roles in cancer. In the TCGA subgroup comparisons for all significantly differentially expressed GOI, the vascular smooth muscle contraction and tight junction pathways highlighted *MYH11* for poor versus good risk, and intermediate versus good risk comparisons. The mineral absorption pathway highlighted *SLC34A2* and *STEAP1* for all three subgroup comparisons while examining all significantly differentially expressed GOI, while the vitamin digestion and absorption pathway highlighted *FOLH1* for the TARGET standard versus low risk and TCGA intermediate versus good risk subgroup comparisons.

## 3. Discussion

We identified 144 GOI as the focus for mRNA-sequence analysis based on literature searches and our own experience of antigen identification. We wanted to examine the expression of these antigens in adult AML in comparison to paediatric patients, an area of research that has had few studies to date. Five tumour associated antigens were found to be differentially expressed between the two subgroups from the TARGET database (standard versus low risk subgroup comparison) and between the three subgroups from the TCGA database (poor versus good, and intermediate versus good risk subgroup comparisons): *SAGE1, MORC4, CEACAM3, SLC34A2* and *STEAP1*. This suggested an overlap in key processes that lead to AML and an impact on the clinical features that lead to the classification of patients into high/poor, standard/intermediate, or low/good risk subgroups. We also examined the survival of patients based on their quartile expression of significantly differentially expressed GOI and determined the pathways involved in each. There were some key differences in the number and type of antigens (between individuals, subgroups and pathways) between the TARGET and TCGA datasets, suggesting that different routes of leukaemia development also exist within these groups.

In adult AML patients, we found significant differences in EFS when comparing intermediate and good (*p* = 0.01), and poor and good risk adult AML patient subgroups (*p* = 0.007). In contrast to paediatric AML patients, differences in EFS were observed when comparing patients with Q1 and Q4 expression levels of *MORC4* (*p* = 0.02) across patients in intermediate and good risk subgroups, and *SAGE1* across patients in poor and good risk subgroups. *MORC4* has been previously identified through the immunoscreening of a testes cDNA library with diffuse large B-cell lymphoma (DLBCL) sera [18]. *MORC4* was subsequently shown to have low level expression in healthy tissues but was shown to be highly expressed in 66% of DLBCL patients [19]. High levels of *MORC4* were demonstrated in healthy placenta and lymph nodes but with little or no expression in myeloid leukaemia cell lines such as K562, HL60 and MOLT-4. The only evidence of a role of *MORC4* in AML to date has been the association between single nucleotide polymorphisms in regions of *MORC4* and the outcomes of patients undergoing autologous stem cell transplantation [20]. *SAGE1* has previously been found to be expressed in 35% of prostate cancers, 33% of oesophageal cancers and 26% of ovarian cancers by qPCR analysis [21], with some limited expression in head and neck cancers. To date, no-one has found an association between *MORC4* and *SAGE1* expression and survival, with this being the first report of this nature.

It was unexpected to find significant differences in *VGLL4* when comparing Q2 and Q3 expression levels across standard and low risk subgroups in the paediatric AML patients, especially because significant differences in *VGLL4* transcription were not found between Q1 and Q4 levels. We had previously described a ‘Goldilocks’ effect of Q1 and Q4 levels of Preferentially Expressed Antigen in Melanoma (*PRAME*) being associated with poor survival in myelodysplastic syndrome patients [22] that we had suggested was caused by high levels of *PRAME* being associated with more aggressive/difficult to treat blast cells and, conversely, low levels of *PRAME* providing insufficient antigen to ignite an immune response, thus both Q1 and Q4 levels of *PRAME* expression were associated with poor survival. However, the reason why Q2 and Q3 levels of *VGLL4* expression is significantly associated with survival in paediatric AML patients may lie in the biological function of *VGLL4*. *VGLL4* was found to be a novel regulator of survival in human embryonic stem cells [23] and acts as a tumour suppressor gene through its interaction with transcriptional enhanced associated domains (TEAD) (reviewed in [24]). It interacts with members of the IAP family, inhibiting their activation and preventing apoptosis, and negatively regulates the Wnt/β-catenin signalling pathway. Lower expression of *VGLL4* is usually associated with poor survival in several cancers, including those affecting the lung, breast, colon, bladder, pancreas and oesophagus; however, none of these findings provide an obvious reason why there was not a significant prognostic difference between patients with Q1 and Q4 levels of *VGLL4,* while there was when levels of *VGLL4* were in the middle quartiles.

One member of the IAP family, *BIRC5* (often called *survivin*), was repeatedly identified throughout this study. As a member of the IAP family of proteins, BIRC5 plays a key role in cellular proliferation and survival. BIRC5 has been shown to be an important antigen in a number of haematological malignancies including adult AML [25], and other investigators have also shown that in adults with AML, BIRC5 levels are higher in bone marrow than paired peripheral blood samples and higher in CD34+CD38− AML blasts than in bulk blasts or total CD34+ cells. Higher levels of BIRC5 are also predictive of a shorter OS and EFS. In this study, we show, for the first time, that there were significant differences in EFS when comparing standard and low risk subgroups of paediatric AML patients (*p* = 0.02) and when comparing patients across both subgroups with the lowest and highest quartile expression levels of *BIRC5* (*p* = 0.03). 

We have previously found the dead box polypeptide 43 (*DDX43*; helicase antigen (*HAGE*)) to be one of the most frequently expressed cancer-testis antigens in adult AML [26] and in this study we found significant differences in *HAGE* expression between intermediate and good risk subgroups of adults with AML (*p* = 7.69 × 10^−3^). This complements previous findings that *HAGE* promoter hypomethylation may be associated with improved OS in AML patients [27] and the finding of an association between high levels of HAGE protein expression and aggressive clinical–pathological features, poor prognosis and worse progression-free survival in breast cancer [28]. 

The cytoplasmic tyrosine kinase *BMX* was significantly different in its expression in paediatric AML between standard and low risk subgroups (*p* = 6.36 × 10^−3^), and in adults when comparing intermediate and good risk subgroups (*p* = 5.74 × 10^−4^). BMX has previously been found to be antigenic through sero-profiling of adult B-ALL [29] and is already used as a target for the small molecule therapy of mature B-cell malignancies [30,31,32] by virtue of its role in the B-cell receptor pathway. The current study suggests that the expression of *BMX* acts as a prognostic indicator in its own right, in adult AML patients (the perceived target of ibrutinib, for whom more than 80% express elevated levels of Bruton’s tyrosinase kinase (BTK) [33]), as well as in paediatric AML patients.

*MYH11* was one of the most significant differentially expressed GOI, differing significantly when comparing intermediate and good risk (*p* = 1.74 × 10^−15^) and poor and good risk adults with AML (*p* = 6.70 × 10^−6^). Indeed, vascular smooth muscle and tight junction pathways were terms that identified *MYH11* as an overlapping KEGG gene in the comparison of these subgroups. *MYH11* is involved in the translocation with core binding factor β (CBFβ) that is typical of 10% of adults with AML [34] and has been shown to target RUNX/ETS-factor binding sites and drive leukaemogenesis through the modulation of H3ac levels [35]. *MYH11* (GKT-ATA18) was also identified as one of the most abundant LAAs when we immunoscreened a testis cDNA library with sera from five adult AML patients with M5 disease [36], but was not preferentially recognised by AML versus healthy donor sera.

Perhaps most notable were not the antigens we had identified prior to this study for further analysis as GOI, but the genes that were significantly differentially expressed and reflected previous findings of the molecular mechanisms underlying AML by other researchers. Significant differences in the expression between risk subgroups of the genes regulated by RUNX3 (formerly known as AML2) were observed through multiple pathways. *RUNX3* was a significant DEG in standard versus low (*p* = 1.21 × 10^−3^), poor versus good (*p* = 1.67 × 10^−14^) and intermediate versus good (*p* = 6.50 × 10^−12^) risk subgroup comparisons, and regulates RUNX1 (formerly known as AML1), a transcription factor that is stabilised through complexing with CBFβ. A loss of function of RUNX1 has been shown to impair the differentiation of both lymphoid and myeloid lineages, often resulting in the development of leukaemia. *RUNX1* is one of the most frequently mutated genes in a variety of haematological malignancies (reviewed most recently by [37]). We found that *RUNX1* was also a significant DEG in standard versus low risk (*p* = 7.78 × 10^−4^) and poor versus good (*p* = 6.12 × 10^−3^) comparisons, as was *RUNX1T1* (standard versus low, *p* = 8.71 × 10^−3^; poor versus good, *p* = 1.28 × 10^−22^; and intermediate versus good risk comparisons, *p* = 2.92 × 10^−17^). RUNX1:RUNX1T1 chimeric protein is a product of the t(8;21) translocation, more commonly known as AML-eight twenty-one (ETO) [38], found in 10% of all de novo AMLs. The translocation has been shown to arrest cell maturation and enable the expansion of stem cells with increased genomic instability [39]. The *AML-ETO* translocation is a favourable prognostic marker associated with higher complete remission rates, OS and progression-free survival [40].

Majeti et al. [41] have previously shown that genes implicated in several pathways, including Wnt signalling, MAP Kinase signalling and adherens junction, were differentially expressed when comparing AML and highly enriched normal stem cells. Our study identified *SSX2IP* in the adherens pathway, with its expression significantly differentiating between intermediate and good risk subgroups of adult AML patients. We had previously identified SSX2IP through the immunoscreening of a testis cDNA library with sera from adult AML patients [36] and it was one of the first tumour antigens shown to act as a biomarker for improved OS when its levels were above median at disease diagnosis in adults with AML who lacked detectable cytogenetic rearrangements [42]. HOX and MLL-driven transformation of haematopoietic stem cells requires Wnt signalling, as AML was shown not to form in the absence of β-catenin otherwise [43] and, in addition to Wnt, a number of HOX and HOX antisense genes were significantly altered in their expression between patient subgroups, including adult poor and good, adult intermediate and good, and paediatric standard and low risk. Caudal-type homeobox transcription factor 2 (CDX2) is capable of upregulating HOX gene expression during embryogenesis and has been found to be upregulated in 90% of AMLs [44]. In addition, SOX genes were found to have significantly different gene expression between risk subgroups. SOX genes share a common HMG domain and all 20 family members are transcription factors that show greater than 60% similarity to the sex-determining region on the Y chromosome (SRY) gene. SOX genes have been shown to be able to act as both oncogenes and tumour suppressor genes in solid tumours (reviewed recently in [45]) and may also be involved in key pathogenetic pathways in AML involving *CEBPA* mutations, activation of β-catenin/Wnt and Hedgehog pathways and aberrant TP53 signals. We found *SOX6* in both TCGA poor versus good and intermediate versus good risk subgroup comparisons and *SOX5*, *SOX15* and *SOX18* in both the TARGET standard versus low risk and TCGA poor versus good risk subgroup comparisons; however, only *SOX18* has previously been implicated in reduced disease-free and OS in AML patients [46].

DNA methyltransferase 3B (DNMT3B) has been shown to play an essential role in the demethylation of transcription factor targets necessary to enable cell differentiation. Lamba et al. [47] recently described the association between increased *DNMT3B* expression and poor clinical outcomes including increased rate of relapse and/or disease resistance in paediatric AML patients. Tumour suppressor genes regulated by DNMT3B have been shown to influence the progression and severity of AML (except *MML-AF9* and inversion of chromosome 16 (inv(16)(p13;q22)). We also found *DNMT3B* to be significantly different in its expression between intermediate and good (*p* = 2.17 × 10^−11^) and poor and good risk (*p* = 2.17 × 10^−11^) subgroups in the TCGA analysis, and standard and low risk (*p* = 4.92 × 10^−3^) subgroups in the TARGET analysis. 

Finally, in terms of the pathways underlying AML, we found both *SLC34A2* and *STEAP1* to be identified within significant KEGG (2016) terms in the standard versus low risk subgroup comparison in the TARGET database, as well as poor versus good risk and intermediate versus good risk comparisons in the TCGA database. In each risk subgroup comparison within both databases, *SLC34A2* and *STEAP1* played a role in mineral absorption. *SLC34A2* was recently identified by sera-screening as being an antigen recognised by sera from patients with immunologically ‘hot’ colorectal cancer [48] and has been shown to play a role in the chemoresistance observed in breast cancer patients [49]. STEAP1 has been shown to be overexpressed in a number of tumour types, while elevated levels of STEAP1 were associated with poor survival in colorectal cancer, DLBCL, multiple myeloma and AML [50]. We had also identified *STEAP1* as an LAA through gene expression analysis of presentation AML patients [51]. Overall, and most significantly, we found that genes altered in AML were involved in key processes such as the evasion of apoptosis (*BIRC5*, *WNT1*) or the control of cell proliferation (*SSX2IP*, *AML1-ETO*), which adds credence to our findings and those of other investigators.

## 4. Materials and Methods

### 4.1. Identification of Antigens and Databases for Investigation

We identified a short-list of 144 tumour antigens based on a comprehensive literature search of currently known tumour antigens in haematological and solid tumours (recently reviewed in [10]). For this study, we obtained paediatric AML data from the TARGET initiative and adult AML data from TCGA. In the TARGET database, EFS was defined as the time from diagnosis until the patient experienced induction failure, relapse or death. In the TCGA database, EFS was defined as the time from diagnosis until relapse or death. OS was defined in both databases as the time from diagnosis until death, or the last date of contact if the patient was still alive. In both databases, a censored status was allocated to patients who did not experience a defined event or were lost to follow-up [52,53].

#### 4.1.1. TARGET Dataset

The results published here are in part based upon data generated by the TARGET initiative (https://ocg.cancer.gov/programs/target) and the AML mRNA-sequencing data we used were specifically derived from phs000465. The full experimental methods used to obtain paired-end mRNA-sequencing data are detailed in [53]. Raw mRNA-sequencing count data and matching clinical data were obtained from the open-access section of the online TARGET Data Matrix for 92 patients from the original TARGET AML cohort. As the majority of the patient samples were sequenced on the HiSeq 2000 (Illumina) with the remaining sequenced on a different model (HiSeq 2500), the decision was made to only include the 65 patients whose samples had been run on the HiSeq 2000 to avoid possible instrument bias. For the purposes of this study, patients were classified as ‘paediatric’ where their age at diagnosis was < 10 years. The age at diagnosis within the TARGET dataset ranged from 137 days (0.4 years) to 3543 days (9.7 years). The highest age reached at final date of patient follow-up was 6357 (17.4 years). Defined by TARGET, patients were assigned a risk group (‘low’, ‘standard’, or ‘high’) based on their level of potential clinical risk arising from cytogenetic or molecular characteristics in relation to AML. Low-risk cytogenetic indicators were the translocation t(8;21)(q22;q22) or the presence of inv(16)(p13q22). Standard risk cytogenetic indicators were *MLL* rearrangements, such as the translocations t(9;11)(q23;p13), t(6;11)(q27;q23), t(11;19)(q23;p13), or t(10;11)(p12;q23), or the presence of a normal karyotype. A molecular characteristic which conferred high clinical risk in normal karyotype patients was the presence of a *FLT3* Internal Tandem Duplication (ITD) [53]. TARGET patient clinical characteristics for standard and low risk subgroups, which were the focus of our analysis, can be viewed in Appendix A.

#### 4.1.2. TCGA Dataset

mRNA-sequencing data from the TCGA-LAML project was used in this analysis. The full experimental methods used to obtain paired-end mRNA-sequencing data on the HiSeq 2000 (Illumina) are detailed in the paper by TCGA Research Network [52]. Raw mRNA-sequencing count data and matching clinical data were available from the Genomic Data Commons (GDC) Data Portal [54] for 151 patients. Patients of FAB classification M3 (APL) and patients possessing the *BCR-ABL1* gene fusion were, however, removed due to their absence in the TARGET AML dataset, thus bringing the total number of patients to 133. For the purposes of this study, patients were classified as ‘adult’ where their age at diagnosis was > 18 years and, within the TCGA dataset, the age at diagnosis ranged from 21 to 88 years. The highest age reached at final date of patient follow-up was 88 years. Defined by TCGA, patients were assigned a risk group (‘good’, ‘intermediate’, or ‘poor’) based on their level of potential clinical risk arising from cytogenetic characteristics and a risk group based on their clinical risk arising from molecular characteristics in relation to AML. Good risk cytogenetic indicators were the translocation t(8;21)(q22;q22) or inv(16)(p13q22); intermediate indicators were the presence of a normal karyotype, the *MLL* translocation t(9;11)(q23;p13), or an intermediate cytogenetic abnormality; poor risk indicators were the presence of complex cytogenetics, a poor risk *MLL* translocation such as t(6;11)(q27;q23) or t(11;19)(q23;p13), or a poor risk cytogenetic abnormality [52]. For the purposes of this study, patients were included in the good, intermediate, or poor risk subgroups only when both cytogenetic and molecular risk classifications matched. TCGA patient clinical characteristics for poor, intermediate and good risk subgroups, which were the focus of our analysis, can be viewed in Appendix A.

### 4.2. DGE Analysis

#### 4.2.1. Prognostic Subgroup Formation

Subgroups were formed on the basis of the genetic risk groups provided within clinical data, and also formed arbitrarily on the basis of the differences in EFS and OS probabilities displayed within Kaplan–Meier curves (where patient numbers allowed). DGE analysis and subsequent survival and pathway analysis were completed using prognostic subgroup pairs, as detailed in Appendix A.

Kaplan–Meier curves of EFS and OS probabilities over time from diagnosis were plotted using the ‘survival’ package [55] within R (version 3.5.2, ‘Eggshell Igloo’) [56]. The same package was used to carry out a log-rank test and assess differences in genetic risk group curves, with a *p* value below 0.05 considered significant. Kaplan–Meier curves also allowed for the selection of arbitrary thresholds which could separate patients within each dataset or cytogenetic group into poor and good prognosis using either EFS or OS. Due to the known risk associated with different primary cytogenetic aberrations, patients with EFS less than 1 year (365 days) and patients with EFS greater than 5 years (1825 days) were selected for those within the inv(16) and t(8;21) cytogenetic groups. Patients with EFS less than 6 months (183 days) and patients with EFS greater than 2 years (730 days) were selected for those possessing *MLL* rearrangements and normal cytogenetics. Patients who were censored within the poor EFS subgroups and patients who were recorded as alive within the poor OS subgroups were removed as they were lost to follow-up within this time period and, therefore, provided no true prognostic information. 

#### 4.2.2. DGE Analysis with DESeq2

Various tools, offering different statistical approaches, exist for DGE analysis and it has previously been demonstrated that no single tool outperforms others in all experimental conditions [57,58]. DESeq2, however, has remained popular due to its stringency and its good balance between specificity and sensitivity. ‘DESeq2’ [59] is a Bioconductor package for R that tests for DGE while considering issues such as large dynamic ranges, discreteness, low numbers of replicates, and the presence of outliers. During DGE analysis with DESeq2, size factors and gene-wise dispersions are estimated automatically, a curve is fitted to the dispersion estimates for each gene, estimates are shrunken towards the expected dispersion values generated by the curve, and the negative binomial model is fitted for each gene.

#### 4.2.3. DESeq2 Workflow

By default, DESeq2 filtered genes with zero counts across all samples, genes with low mean normalised counts (via its independent filtering function), and genes with extreme count outliers (assessed using Cook’s distance) when more than three replicates were present [59]. Although unnecessary for analysis, genes with less than a total of ten counts across all samples were pre-filtered to reduce run time. 

DESeq2’s default settings were used while performing DGE analysis with the ‘DESeq’ function, which involves the estimation of size factors and dispersions, the fitting of the negative binomial model, and the calculation of Wald statistics. While generating results using DESeq2’s ‘results’ function, the contrast argument was used to specify the comparison and reference factors, the alpha argument was specified as the intended adjusted *p* value threshold (0.01), the BH adjustment method was specified, and independent filtering was implemented to optimise the number of significant genes by using the mean of normalised counts as a filter statistic. The DESeq2 function ‘lfcshrink’ was used to shrink log2 fold change estimates toward zero for genes with low counts or high dispersion values following the generation of results [59]. Gene annotation was carried out using the ‘AnnotationDbi’ [60] and ‘org.Hs.eg.db’ [61] Bioconductor R packages. The results of this analysis for significant genes (adjusted *p* value < 0.01) were ordered by increasing adjusted *p* value before being combined with normalised (by size factor) counts. 

Significantly DEGs discovered by DGE analysis were compared to the list of GOI provided and the results for any overlapping genes were selected and exported. Volcano plots were generated (using R’s generic ‘plot’ function along with the aid of the ‘calibrate’ package [62]) to illustrate all significantly DEGs (adjusted *p* value < 0.01) along with their log2 fold changes, and to highlight any significantly differentially expressed GOI. No further downstream analysis was undertaken where no significantly differentially expressed GOI were discovered. 

### 4.3. Survival Analysis

A regularised log transformation (using DESeq2’s ‘rlogTransformation’ function, not blind to the specified DESeqDataSet object design formula) was implemented on the raw count data for significantly differentially expressed GOI [59]. These transformed expression counts were assigned to a quartile (Q1/Q2/Q3/Q4) depending on the thresholds for each gene. Both the transformed values and quartile values were exported. The ‘survdiff’ function within the ‘survival’ package was used to perform log-rank tests for each gene to compare EFS of patients with expression levels in Q1 to those with expression in Q4. Further log-rank tests were performed to compare EFS of patients with expression levels within Q2 to Q3. The results with a *p* value less than 0.05 were deemed significant. The ‘survutils’ R package [63] was used to isolate the *p* values from these tests. If either log-rank test result (Q1–Q4 or Q2–Q3) was significant, a Kaplan–Meier plot was generated for that gene to illustrate the EFS probability of patients with expression levels in each quartile over time. 

### 4.4. Pathway Analysis

Pathway analysis was performed on total, downregulated (log2 fold change < 0), and upregulated (log2 fold change > 0) significantly differentially expressed GOI. The ‘enrichR’ package [64] was used to provide an interface to the Enrichr KEGG (2016) database within R [65,66]. The results were ordered according to Enrichr’s combined score (log of the *p* value outcome of Fisher’s exact test multiplied by the Z-score of the deviation from the expected rank [65]). 

## 5. Conclusions

Our study shows that paediatric and adult AML patients express, perhaps not surprisingly, some similar and some very different antigens with regards to gene expression and biomarkers for EFS. AML blasts from paediatric patients appear to have a larger range of dysregulated antigens suggesting a mechanism by which their hosts may achieve improved outcome following chemotherapy. We propose that the aberrant expression of a wider range of antigens by AML blasts in paediatric patients may facilitate an effective immune response that more easily recognises these blasts as different from healthy haematopoietic stem cells. For the first time, we have provided unique insights into the differences and similarities in the gene expression that underlies AML, providing new insights into the significantly different gene expression between risk subgroups and the molecular pathways involved in each.

## Figures and Tables

**Figure 1 cancers-12-02769-f001:**
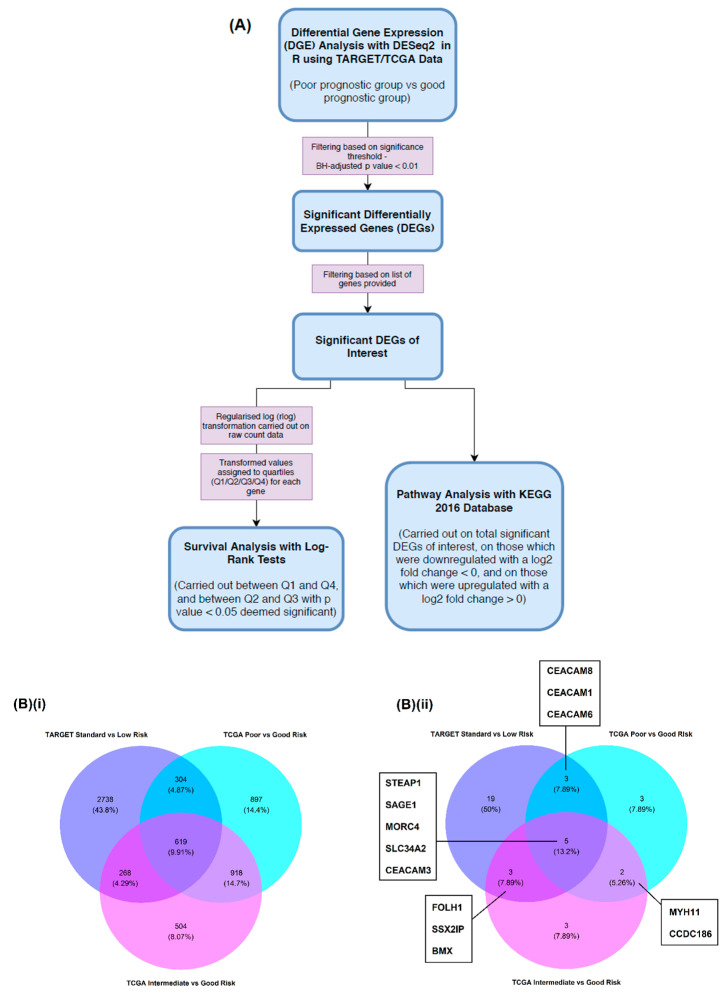
Overview of the analytical workflow and identification of differentially expressed genes (DEGs) that overlap between subgroup comparisons. (**A**) DESeq2 was used to perform differential gene expression (DGE) analysis using mRNA-Seq raw counts for various poor prognostic versus good prognostic group comparisons within Therapeutically Applicable Research to Generate Effective Treatments (TARGET), and The Cancer Genome Atlas (TCGA) patient datasets. DGE results were filtered (Benjamini–Hochberg (BH)-adjusted *p* value < 0.01) to yield significant DEGs. Where genes of interest (GOI) were found to be significantly differentially expressed, raw count data within a corresponding subgroup comparison was transformed and expression levels of GOI across all patients in the comparison were assigned to a quartile (Q1/Q2/Q3/Q4). Log-rank tests were carried out between patients of Q1 and Q4 expression, and of Q2 and Q3, with *p* value < 0.05 deemed significant. Pathway analysis was also carried out using the Kyoto Encyclopedia of Genes and Genomes (KEGG) 2016 Database on total significantly differentially expressed GOI, on those considered as downregulated (log2 fold change < 0), and those considered as upregulated (log2 fold change > 0); (**B**) Venn diagrams provide a visual summary of (**i**) all genes and (**ii**) GOI that were identified as significantly differentially expressed when comparing TARGET and TCGA subgroups.

**Figure 2 cancers-12-02769-f002:**
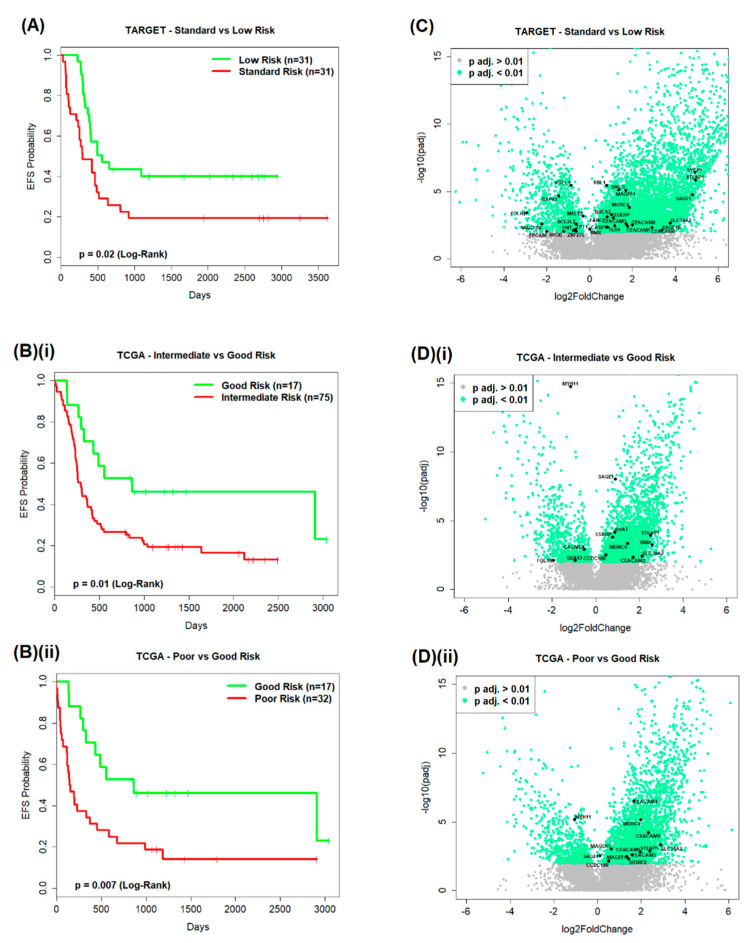
DGE analysis. Pairwise comparisons of event-free survival (EFS) of (**A**) TARGET patients by standard versus low risk (*p* = 0.02), and (**B**) TCGA patients by (**i**) intermediate versus good risk (*p* = 0.01) and (**ii**) poor versus good risk (*p* = 0.007) subgroups. All *p* values derived from log-rank analysis. Volcano plots represent the genes that were differentially expressed by *p* > 0.01 (grey dots) or *p* < 0.01 (green dots) when comparing (**C**) TARGET patients by standard versus low risk, and (**D**) TCGA patients by (**i**) intermediate versus good and (**ii**) poor versus good risk subgroups. Names and black dots indicate the position of antigens listed as GOI when studying gene expression.

**Figure 3 cancers-12-02769-f003:**
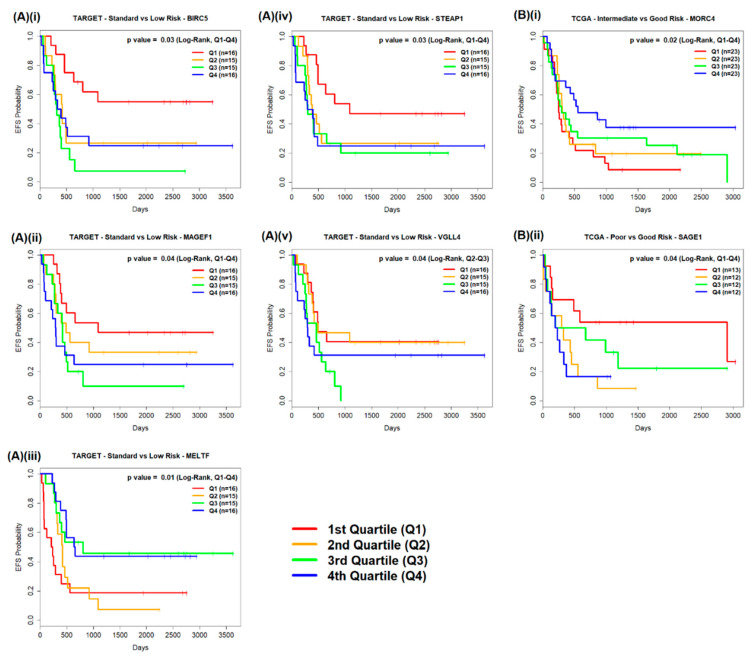
Significant impacts of gene expression levels (divided into quartiles) on patient EFS. (**A**) TARGET study patients across the standard and low risk subgroups were found to have significant differences in EFS when comparing quartile expression levels of (**i**) *BIRC5*; (**ii**) *MAGEF1*; (**iii**) *MELTF*; (**iv**) *STEAP1* and (**v**) *VGLL4*; (**B**) TCGA study patients across the intermediate and good risk subgroups showed significant differences in EFS when comparing quartile expression levels of (**i**) *MORC4*, and patients across poor and good risk subgroups showed significant differences in EFS when comparing quartile expression levels of (**ii**) *SAGE1*.

**Table 1 cancers-12-02769-t001:** Genes of Interest (GOI) and other notable genes identified as significantly differentially expressed (BH-adjusted *p* < 0.01) between (**A**) standard and low risk subgroups within the TARGET dataset, (**B**) intermediate and good risk subgroups within the TCGA dataset, and (**C**) poor and good risk subgroups within the TCGA dataset.

(A)	Gene Symbol	Log2 Fold Change	BH-Adjusted *p* Value	(B)	Gene Symbol	Log2 Fold Change	BH-Adjusted *p* Value	(C)	Gene Symbol	Log2 Fold Change	BH-Adjusted *p* Value
**GOI**	BCL2L2	−0.63	2.61 × 10^−3^	**GOI**	BMX	2.57	5.74 × 10^−4^	**GOI**	CCDC186	0.51	7.05 × 10^−3^
	BIRC5	1.35	7.48 × 10^−6^		CALML4	−0.53	1.25 × 10^−3^		CEACAM1	1.67	3.10 × 10^−7^
	BMX	0.01	6.36 × 10^−3^		CCDC186	0.46	2.96 × 10^−3^		CEACAM3	1.59	2.36 × 10^−3^
	BRCA2	1.02	5.12 × 10^−2^		CEACAM3	1.71	4.33 × 10^−3^		CEACAM6	1.96	1.57 × 10^−3^
	CASP1	0.84	4.31 × 10^−3^		DDX43	−0.95	7.69 × 10^−3^		CEACAM8	2.33	5.66 × 10^−5^
	CEACAM1	1.76	3.89 × 10^−3^		FOLH1	−1.94	7.52 × 10^−3^		MAGEE1	1.33	3.34 × 10^−3^
	CEACAM3	1.70	2.57 × 10^−3^		HHAT	0.87	6.75 × 10^−5^		MAGEH1	0.62	8.86 × 10^−4^
	CEACAM6	2.91	4.80 × 10^−3^		MORC4	1.45	4.40 × 10^−4^		MORC4	1.98	7.20 × 10^−6^
	CEACAM8	1.99	3.13 × 10^−3^		MYH11	−1.15	1.74 × 10^−15^		MYH11	−1.05	6.70 × 10^−6^
	DAPK1	−1.45	2.12 × 10^−5^		SAGE1	0.89	9.64 × 10^−9^		SAGE1	0.11	2.85 × 10^−3^
	EPCAM	−2.00	9.72 × 10^−3^		SLC34A2	2.12	3.72 × 10^−3^		SH3RF2	1.42	4.72 × 10^−3^
	FANCC	0.81	7.45 × 10^−4^		SSX2IP	0.77	1.55 × 10^−4^		SLC34A2	2.89	4.65 × 10^−4^
	FHIT	−0.76	7.58 × 10^−3^		STEAP1	2.51	1.19 × 10^−4^		STEAP1	2.38	1.29 × 10^−3^
	FOLH1	−2.95	3.97 × 10^−4^	**Other Notable Genes**	CD52	−1.60	1.31 × 10^−4^	**Other Notable Genes**	CD19	−1.92	2.52 × 10^−6^
	MAGED4	−2.24	2.65 × 10^−3^		DNMT3B	1.95	2.17 × 10^−11^		CD276	2.19	1.67 × 10^−4^
	MELTF	−0.30	6.72 × 10^−4^		HMNT	2.99	5.48 × 10^−10^		CD7	1.97	7.27 × 10^−6^
	MORC4	1.86	1.54 × 10^−4^		HOXA10	3.55	1.59 × 10^−13^		CD70	2.32	1.80 × 10^−4^
	RBL1	0.80	3.51 × 10^−6^		HOXA10-AS	3.54	8.13 × 10^−13^		CD81	1.33	1.67 × 10^−5^
	RHOB	−1.22	9.32 × 10^−3^		HOXA11	2.70	1.10 × 10^−12^		CD82	1.82	6.60 × 10^−11^
	SAGE1	4.80	1.78 × 10^−5^		HOXA11-AS	2.25	1.96 × 10^−11^		DNMT3B	2.62	3.94 × 10^−14^
	SLC34A2	3.75	2.20 × 10^−3^		HOXA13	2.70	4.16 × 10^−8^		HOXA1	1.33	8.85 × 10^−4^
	SSX2IP	1.11	8.69 × 10^−4^		HOXA2	3.34	6.09 × 10^−15^		HOXA10	3.42	1.18 × 10^−9^
	STEAP1	4.95	1.37 × 10^−6^		HOXA3	5.30	6.50 × 10^−33^		HOXA10-AS	3.58	7.04 × 10^−11^
	SYCP1	4.91	3.92 × 10^−7^		HOXA4	4.48	1.50 × 10^−21^		HOXA11	3.09	3.10 × 10^−8^
	TLR4	1.16	3.56 × 10^−3^		HOXA5	4.58	7.43 × 10^−22^		HOXA11-AS	2.30	1.52 × 10^−5^
	TPT1	−0.65	6.73 × 10^−3^		HOXA6	5.83	1.04 × 10^−34^		HOXA2	2.91	7.86 × 10^−11^
	VGLL4	−0.86	3.43 × 10^−6^		HOXA7	4.92	1.19 × 10^−24^		HOXA3	5.08	3.57 × 10^−26^
	XAGE1B	3.37	7.41 × 10^−3^		HOXA9	4.19	6.04 × 10^−18^		HOXA4	3.94	3.99 × 10^−14^
	ZNF275	−0.64	9.11 × 10^−3^		HOXA-AS2	5.39	6.77 × 10^−35^		HOXA5	4.19	1.69 × 10^−14^
**Other Notable Genes**	CD34	−2.91	1.53 × 10^−7^		HOXA-AS3	6.25	1.30 × 10^−39^		HOXA6	6.17	7.87 × 10^−32^
	CD52	−2.68	2.79 × 10^−12^		HOXB1	1.86	5.60 × 10^−3^		HOXA7	4.59	5.05 × 10^−16^
	CD99	−1.51	2.48 × 10^−8^		HOXB3	2.39	1.22 × 10^−6^		HOXA9	4.21	1.67 × 10^−14^
	DNMT3B	1.13	4.92 × 10^−3^		HOXB4	1.83	3.63 × 10^−4^		HOXA-AS2	5.07	9.73 × 10^−26^
	HOXA10	3.29	4.57 × 10^−9^		HOXB5	4.66	5.13 × 10^−22^		HOXA-AS3	6.31	3.19 × 10^−33^
	HOXA10-AS	2.69	3.65 × 10^−4^		HOXB6	4.60	1.48 × 10^−21^		HOXB5	3.43	3.50 × 10^−7^
	HOXA7	3.90	2.05 × 10^−7^		HOXB7	3.69	2.53 × 10^−13^		HOXB6	3.50	6.34 × 10^−4^
	HOXA-AS3	6.41	3.79 × 10^−19^		HOXB8	4.49	2.09 × 10^−12^		HOXB-AS3	4.06	1.02 × 10^−5^
	HOXB7	3.35	1.13 × 10^−10^		HOXB9	4.73	1.48 × 10^−13^		HOXC4	1.74	2.15 × 10^−3^
	HOXB-AS3	1.22	2.20 × 10^−12^		HOXB-AS3	5.12	3.85 × 10^−27^		IL10	1.61	6.55 × 10^−7^
	IL11	−2.55	6.73 × 10^−4^		IL10	0.41	9.84 × 10^−3^		IL11	1.40	1.30 × 10^−4^
	IL15	1.98	4.86 × 10^−4^		IL15	2.28	1.15 × 10^−7^		IL15	1.85	1.21 × 10^−5^
	IL19	3.18	1.45 × 10^−4^		IL4	−1.24	1.36 × 10^−3^		IL7	4.23	3.14 × 10^−21^
	IL24	4.56	3.41 × 10^−5^		IL7	2.93	5.27 × 10^−7^		PRICKLE1	1.55	3.78 × 10^−4^
	IL3	−2.92	7.12 × 10^−3^		PRICKLE1	2.16	5.06 × 10^−9^		PRICKLE4	1.16	3.71 × 10^−3^
	IL7	2.63	3.73 × 10^−5^		PRICKLE2	2.32	1.56 × 10^−5^		RUNX1	−0.48	6.12 × 10^−3^
	RUNX1	−0.98	7.78 × 10^−4^		RUNX1T1	−5.72	2.92 × 10^−17^		RUNX1T1	−6.70	1.28 × 10^−22^
	RUNX3	1.24	1.21 × 10^−3^		RUNX3	1.98	6.50 × 10^−12^		RUNX3	2.50	1.67 × 10^−14^
	SOX11	6.43	2.45 × 10^−8^		SOX6	1.17	6.86 × 10^−4^		SOX15	−0.68	7.94 × 10^−3^
	SOX15	−2.65	7.94 × 10^−9^		TRIM10	2.47	2.77 × 10^−7^		SOX18	2.17	2.01 × 10^−4^
	SOX18	−2.21	4.93 × 10^−4^		TRIM15	2.09	2.87 × 10^−4^		SOX5	2.50	1.65 × 10^−3^
	SOX30	2.56	3.86 × 10^−4^		TRIM24	−0.54	3.49 × 10^−3^		SOX6	1.66	1.61 × 10^−5^
	SOX5	2.26	6.16 × 10^−5^		TRIM29	2.96	1.20 × 10^−7^		TOX2	2.11	5.61 × 10^−6^
	SOX8	−1.99	3.69 × 10^−3^		TRIM47	−2.08	7.03 × 10^−10^		TRIM10	2.91	4.65 × 10^−8^
	TET1	−1.22	1.08 × 10^−3^		TRIM6	−1.11	5.88 × 10^−4^		TRIM24	−0.76	1.34 × 10^−6^
	TRIM10	0.67	4.54 × 10^−4^		TRIM68	0.42	5.52 × 10^−3^		TRIM29	3.77	6.26 × 10^−6^
	TRIM24	−1.16	2.33 × 10^−5^		TRIM71	−2.01	3.94 × 10^−5^		TRIM36	−1.50	4.51 × 10^−3^
	TRIM29	1.68	3.33 × 10^−6^		TRIM8	0.88	8.27 × 10^−5^		TRIM40	2.23	4.01 × 10^−3^
	TRIM31	3.20	3.33 × 10^−3^		TRIM9	2.72	7.94 × 10^−13^		TRIM72	−1.25	3.32 × 10^−3^
	TRIM35	0.59	4.38 × 10^−3^						TRIM8	1.06	1.55 × 10^−4^
	TRIM38	0.60	3.51 × 10^−3^						TRIM9	2.68	2.00 × 10^−9^
	TRIM47	−2.65	3.25 × 10^−9^						WNT6	3.14	3.99 × 10^−10^
	TRIM59	0.95	1.64 × 10^−3^								
	TRIM61	2.30	5.56 × 10^−3^								
	TRIM69	0.67	9.44 × 10^−3^								
	TRIM7	2.46	3.84 × 10^−6^								
	TRIM71	−3.09	4.81 × 10^−7^								
	TRIM8	0.75	2.92 × 10^−3^								
	TRIM9	2.49	2.82 × 10^−6^								

**Table 2 cancers-12-02769-t002:** Significant log-rank test results for significantly differentially expressed GOI.

	Patient Subgroup Comparison	Significant DEGs of Interest	*n* of Q1, Q4	Q1–Q4 Log-Rank *p* Value	Q1 Median EFS Survival in Days (95% CL)	Q4 Median EFS Survival in Days (95% CL)	*n* of Q2, Q3	Q2-Q3 Log-Rank *p* Value	Q2 Median EFS Survival in Days (95% CL)	Q3 Median EFS Survival in Days (95% CL)
**TARGET**	**Standard Risk vs. Low Risk** (*n* = 31 vs. *n* = 31)	*BIRC5*	16, 16	0.0293	NR (637–NA)	362 (237–NA)	15, 15	0.2169	419 (291–NA)	321 (269–655)
		*MAGEF1*	16, 16	0.0376	1093 (404–NA)	292 (102–NA)	15, 15	0.2338	488 (294–NA)	419 (269–NA)
		*MELTF*	16, 16	0.0127	225 (71–NA)	646 (488 – NA)	15, 15	0.0849	419 (321–1093)	809 (366–NA)
		*STEAP1*	16, 16	0.0305	1093 (497–NA)	344 (77–NA)	15, 15	0.5139	383 (321–NA)	299 (269–NA)
		*VGLL4*	16, 16	0.2087	497 (395–NA)	296 (109–NA)	15, 15	0.0368	496 (383–NA)	461 (269–NA)
**TCGA**	**Intermediate Risk vs. Good Risk** (*n* = 75 vs. *n* = 17)	*MORC4*	23, 23	0.0235	253 (219–517)	554 (362–NA)	23, 23	0.8741	310 (259–834)	298 (234–2121)
	**Poor Risk vs. Good Risk** (*n* = 32 vs. *n* = 17)	*SAGE1*	13, 12	0.0394	2910 (158–NA)	216 (113–NA)	12, 12	0.2502	314 (119–NA)	438 (128–NA)

NR = not reached; NA = not available.

**Table 3 cancers-12-02769-t003:** Pathway analysis using the most common KEGG terms overall between both TARGET and TCGA data when examining the (**A**) total, (**B**) downregulated (log2 FC < 0), and (**C**) upregulated (log2 FC > 0) significantly differentially expressed GOI identified from subgroup comparisons (with > 10 patients per subgroup) during DGE analysis.

	KEGG (2016) Term	Overlapping KEGG (2016) Genes	No. of Subgroup Comparisons
(**A**)	Mineral absorption Homo sapiens_hsa04978	*SLC34A2, STEAP1*	3
	Vitamin digestion and absorption_Homo sapiens_hsa04977	*FOLH1*	2
(**B**)	Alanine, aspartate and glutamate metabolism_Homo sapiens_hsa00250	*FOLH1*	2
	Vitamin digestion and absorption_Homo sapiences_hsa04977	*FOLH1*	2
(**C**)	Mineral absorption_Homo sapiens_hsa04978	*SLC34A2, STEAP1*	3

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
