# Peer review of "Identification of Genes Whose Expression Overlaps Age Boundaries and Correlates with Risk Groups in Paediatric and Adult Acute Myeloid Leukaemia"

_cancers, 2020, doi:10.3390/cancers12102769_

Round 1
Reviewer 1 Report
Nothing to add to the previous revision.
Reviewer 2 Report
None
Reviewer 3 Report
The authors satisfied my observations.
Reviewer 4 Report
All of my comments have been properly addressed.
This manuscript is a resubmission of an earlier submission. The following is a list of the peer review reports and author responses from that submission.
Round 1
Reviewer 1 Report
The manuscript entitled “Identification of genes whose expression overlaps age boundaries and predicts risk groups in paediatric/adolescent and adult acute myeloid leukaemia” by Davis et al. deals with a research topic that is interesting for clinicians and translational researchers. The authors utilized two publicly available data sets (TARGET and TCGA) to search for genes that are differentially expressed between standard and low risk pediatric AML and poor and good risk adult AML patients’ subgroups. Additionally, the authors identified genes that have a prognostic impact in each patient cohort. Although this manuscript provides important information, there are a number of issues that needs to be addressed.
Major comments:
1) The author should provide additional information on the definitions that they used. What is the cut-off age between children/adolescence and adults? What does “good”, “poor”, “standard risk”, “low risk” and “intermediate risk” mean?
2) The authors correctly mention that AML is a very heterogeneous disease. I think it is problematic that the authors combined all adult AML patients, especially for the outcome analyses. For example, several studies have previously shown that important differences can be found between younger adult AML patients and older AML patients (>60 years). Especially, since those patients get different treatments. Subgroup analyses for age groups and cytogenetic subgroups would further strengthen the manuscript.
3) As mentioned above, the used risk classification is not well defined. Maybe the authors should also use the very well established ELN risk groups for their analyses?
4) The authors should try to better explain and define, how they chose the list of 144 genes of interested. Also, statements like “notable genes” are difficult to follow. Why were these genes considered “notable”?
5) The authors introduce eight (for TARGET) and five (TCGA) prognostic subgroups. It remains unclear how these groups have been chosen?
6) Since the authors performed several outcome analyses, I think it is important to provide more clinical information of the patients. How have the patients been treated and consolidated? How many have received an allogeneic stem cell transplantation?
7) Several of the genes that were found by the authors have not been well described as prognosticates in AML yet, which is very interesting. However, I think it would be very important to validate these findings e.g. in independent patients’ cohort.
Minor comments:
1) The labeling of Figure 2 is confusing. I suggest using the letters A-E for the figure.
2) The number of patients in each group should be given in Figure 3 for each curve. Alternatively, the number of patients at risk can be provided.
3) The Introduction and Discussion are quite long, especially compared to the Results. Maybe shortening these sections can help to focus on the important findings.
Author Response
Responses to the reviewers’ comments
Reviewer 1 - Major Comments:
1) The author should provide additional information on the definitions that they used. What is the cut-off age between children/adolescence and adults? What does “good”, “poor”, “standard risk”, “low risk” and “intermediate risk” mean?
> Age cut-offs have been provided to indicate how we have defined paediatric and adults. These follow the criteria used by the TARGET/TCGA and are detailed on lines 451-454 and 470-472, respectively.
> In addition we used TARGET/TCGA risk groups that are based on the genetics within the samples – both cytogenetic and molecular aberrations. This is now clarified in lines 454-462 and 472-483 and a new supplementary table 3 with patient clinical info is now provided.
2) The authors correctly mention that AML is a very heterogeneous disease. I think it is problematic that the authors combined all adult AML patients, especially for the outcome analyses. For example, several studies have previously shown that important differences can be found between younger adult AML patients and older AML patients (>60 years). Especially, since those patients get different treatments. Subgroup analyses for age groups and cytogenetic subgroups would further strengthen the manuscript.
> This point is well taken and we hope it will be the focus of future work that would need to combine multiple RNA-seq studies, if/when they become available, to get the numbers needed to do this analysis justice. The current study focused on risk subgroups (but was mindful of a number of clinical factors that influence them).
3) As mentioned above, the used risk classification is not well defined. Maybe the authors should also use the very well established ELN risk groups for their analyses?
> We followed the risk classifications defined as part of the published TCGA and TARGET studies, and these are now clearly explained in lines 454-462 and 472-483. This seemed more appropriate than trying to apply risk group criteria to patients whose full clinical notes we were not privy to.
4) The authors should try to better explain and define, how they chose the list of 144 genes of interested. Also, statements like “notable genes” are difficult to follow. Why were these genes considered “notable”?
> GOI were chosen based on a comprehensive literature review and this is now clarified on lines 434-435.
5) The authors introduce eight (for TARGET) and five (TCGA) prognostic subgroups. It remains unclear how these groups have been chosen?
> Risk subgroups were chosen following the genetic risk classifications provided by TARGET/TCGA. This is better clarified on lines 454-462 and 472-483.
6) Since the authors performed several outcome analyses, I think it is important to provide more clinical information of the patients. How have the patients been treated and consolidated? How many have received an allogeneic stem cell transplantation?
> This information is provided as supplementary material in Supplementary Figure 1 and Supplementary Table 3.
7) Several of the genes that were found by the authors have not been well described as prognosticates in AML yet, which is very interesting. However, I think it would be very important to validate these findings e.g. in independent patients’ cohort.
> In this study we were identifying genes that were differentially expressed between risk subgroups in samples from adult and paediatric AML patients. It would be interesting to further characterise the protein expression of these genes in AML samples, using appropriate subsets of bone marrow cells as normal donor controls. However normal donor control cell populations were not analysed in the TCGA or TARGET studies making it almost impossible to judge the value of these genes and their products as prognosticates.
Minor Comments:
1) The labeling of Figure 2 is confusing. I suggest using the letters A-E for the figure.
> Letters have been provided beside each Kaplan-Meier/volcano plot within Figure 2 to avoid confusion.
2) The number of patients in each group should be given in Figure 3 for each curve. Alternatively, the number of patients at risk can be provided.
> The numbers of patients in each quartile are now indicated in parenthesis in the key (line 288).
3) The Introduction and Discussion are quite long, especially compared to the Results. Maybe shortening these sections can help to focus on the important findings.
> The Introduction is not too long (51 lines) but we agree the discussion did seem disproportionately long. We have shortened the Discussion (by 40 lines and 16 references) and added information requested by the reviewers to the other sections, which we agree brings a better balance the paper and facilitates the focus on the important findings.
Reviewer 2 Report
Davis et al. proposed a comparative study on AML patient’s mRNA-sequencing data from two publicly databases: NCI-TARGET and TCGA. Patients selected from TARGET database belongs to children and adult children instead data from TCGA belongs to adult patients. First, authors drafted a list of Genes Of Interest (GOI) from literature searches representing tumors antigens both in hematological and solid tumors.
For each patient group different subgroups where detected based on the of the event free survival and differential gene expression analysis where performed between them. The significant differentially expressed genes where then compared with the GOI to detect differentially expressed tumors antigens. The statistical relationship between differentially expressed GOI, and EFS has been then analyzed and some correlated genes were detected in both groups. Finally, pathway analysis has been performed on differentially expressed GOIs in order to frame them in a functional context.
In the race for the detection of targets for immunotherapies this study offers valuable starting to select antigens to tests in experimentally other than offering new understandings in AML pathogenesis.
I recommend for its publication with some minor revision:
- Figure 1B should be cited in the main text maybe in the paragraph 2.1 and should be reformatted (it results incomplete on the left).
- In the paragraph 2.2 a reference to the complete figure 3 should be included.
- Doi for zenodo are not searchable from the search engine maybe could be easier for the readers to use relative link (e.g. use https://doi.org/10.5281/zenodo.3857773 instead of DOI: 10.5281/zenodo.3857773).
Author Response
Rev 2 – Minor Comments:
1) Figure 1B should be cited in the main text maybe in the paragraph 2.1 and should be reformatted (it
results incomplete on the left).
> Figure 1B is now cited on lines 91 and 92.
2) In the paragraph 2.2 a reference to the complete figure 3 should be included.
> Figure 3 is referred to on lines 222 and 223
3) DOI for zenodo are not searchable from the search engine maybe could be easier for the readers to use relative link (e.g. use https://doi.org/10.5281/zenodo.3857773 instead of DOI: 10.5281/zenodo.3857773).
> These have been corrected on lines 569 and 570. Table S2 was removed in response to reviewer 1’s comment, another table and a figure added, all similarly referenced in terms of DOI format.
Reviewer 3 Report
In this manuscript dr. Davis and coworkers analyzed mRNA-sequencing data from two publicly accessible databases, NCI-TARGET and TCGA, deriving from pediatric/adolescent and adult AML, respectively.
The authors identified genes differentially expressed between risk patient subgroups within pediatric and adult AML.
They found differences in event-free survival when comparing risk subgroups and quartile expression levels of genes differently expressed within each cohort.
They also examined KEGG pathway data and found that genes differentially expressed in AML are involved in key processes such as the evasion of apoptosis or the control of cell proliferation.
General comment:
The authors provide an interesting re-analysis of public available m-RNA expression data in AML that could potentially be useful in clinical approach to AML patients. Unfortunately, the complexity of these scientific data makes the readability of the manuscript not simple.
Major:
1-Lane 3: Author constructed prognostic groups based on biological/survival data and subsequently searched for differentially expressed genes, therefore the title should down toned: I suggest substitute “predicts” with “correlates with” or “associates with”.
2-Following this point authors should describe in more detail how they constructed the comparison groups, which clinical and biological characteristics they considered, how they used EFS and OS information for subgroup creation and outcome evaluation and how they defined cut-offs.
More details are required for the criteria used to censor patients without events (lane from 492 to 495) to exclude it could represent a selection bias.
These additional informations can be provided in the manuscript or as supplemental material and methods.
3-Lane 20: as stated in the material and method section, the cohort derived from the TARGET cohort has an age at diagnosis ranging from 113 days (0.3 years) and 3543 days (9.7 years). Therefore it seems more appropriate to refer to this cohort as pediatric only through the whole text.
4-Lane 314 to 329: Some genes and genes’ data cited in the discussion can be only found in figures and supplementary tables. It would be appropriate to implement in the result section a short paragraph and a table summarizing gene and gene data considered of interest (i.e. genes highlighted in figure2 panels (C), (D)(i) and (ii)) and/or selected for further discussion. The table should include gene symbol, log2Fold Change, p adj., cohort of data considered (TARGET/TCGA), patients subgroup in which this comparison is done, ...
Minor:
Lane 102 (paragraph 2.1.1): the material and method section are at and of the manuscript, therefore it is important to anticipate that the TARGET cohort is pediatric
Lane 110 to 116: revise the syntax of the period
Figure 1: for panel B the label (B) (i) are missing
Lane 149 (paragraph 2.1.2): the material and method section are at and of the manuscript, therefore it is important to anticipate that the TCGA AML data are from adult patients
Lane 203: verify the number (twelve) of the significant DEGs of interest
Lane 216: the material and method section are at and of the manuscript, therefore briefly define what KEGG is
Lane 378: check the spelling of c-EBPa
Author Response
Reviewer 3 – Major Comments:
1) Lane 3: Author constructed prognostic groups based on biological/survival data and subsequently searched for differentially expressed genes, therefore the title should down toned: I suggest substitute “predicts” with “correlates with” or “associates with”.
> Title changed as suggested.
2) Following this point authors should describe in more detail how they constructed the comparison groups, which clinical and biological characteristics they considered, how they used EFS and OS information for subgroup creation and outcome evaluation and how they defined cut-offs.
More details are required for the criteria used to censor patients without events (lane from 492 to 495) to exclude it could represent a selection bias.
These additional informations can be provided in the manuscript or as supplemental material and methods.
> TARGET/TCGA definitions were used to choose patients for inclusion and construct risk subgroups.
> EFS and OS are defined within the manuscript (lines 437-441). Cytogenetic classifications (inv(16), t(8;21), MLL abnormality, normal karyotype..) are defined to indicate how comparisons were performed where patient numbers allowed.
3) Lane 20: as stated in the material and method section, the cohort derived from the TARGET cohort has an age at diagnosis ranging from 113 days (0.3 years) and 3543 days (9.7 years). Therefore it seems more appropriate to refer to this cohort as pediatric only through the whole text.
> All ‘paediatric/adolescent’ references have been changed to read ‘paediatric’.
4) Lane 314 to 329: Some genes and genes’ data cited in the discussion can be only found in figures and supplementary tables. It would be appropriate to implement in the result section a short paragraph and a table summarizing gene and gene data considered of interest (i.e. genes highlighted in figure2 panels (C), (D)(i) and (ii)) and/or selected for further discussion. The table should include gene symbol, log2Fold Change, p adj., cohort of data considered (TARGET/TCGA), patients subgroup in which this comparison is done, ...
> Added new table within the paper (Table 1, line 164).
Minor Comments:
1) Lane 102 (paragraph 2.1.1): the material and method section are at and of the manuscript, therefore it is important to anticipate that the TARGET cohort is pediatric.
> This is now explained in the Introduction (lines 78-85) and added to the abstract (lines 16 and 17).
2) Lane 110 to 116: revise the syntax of the period
> Sorry, we considered how else to write the sentence but were at a loss.
3) Figure 1: for panel B the label (B) (i) are missing.
> Now visible.
4) Lane 149 (paragraph 2.1.2): the material and method section are at and of the manuscript, therefore it is important to anticipate that the TCGA AML data are from adult patients
> This is now explained in the Introduction (lines 78-85) and added to the abstract (lines 16 and 17).
5) Lane 203: verify the number (twelve) of the significant DEGs of interest
> Number corrected to five (line 225).
6) Lane 216: the material and method section are at and of the manuscript, therefore briefly define what KEGG is
> Definition inserted into the abstract (line 25) and definition remains in the Figure 1 legend where it first appears in the main body of the manuscript.
7) Lane 378: check the spelling of c-EBPa
> Corrected on line 400 and throughout.
Reviewer 4 Report
The manuscript gave an interesting overview of gene expression in relation to prognosis in AML of different ages. The following criticisms should be addressed
Major points:
- GROUP DEFINITION: Subgroups were formed for the comparisons using genetic characteristics (which ones?) and on the basis of survival (material and methods, line 478-480). Why the same groups were analyzed for EFS (results, line 93-97) as reported in the results? EFS is a value used for group definition or a variable to evaluate?
- COMPARISON: the authors asserted that gene expression differences were examined in a cohort of adult AML in comparison to pediatric/adolescent AML. However the comparison was carried out among different subgroups in adult database and among different subgroups in pediatric/adolescent database, but a direct comparison between adult pediatric/adolescent was not described.
Minor points:
- The introduction section is too long. It should be shortened and help the common reader to better understand this difficult analysis.
- Both in paragraphs 2.1.1 and 2.1.2, the authors listed the prognostic subgroups in which the analysis did not allow to continue a focused analysis on differential expressed GOI. Nevertheless the 3 subgroups identified for further analysis were not explicitly specified. A specification could help in reading results.
- Figure 1: the letter “B” and (i) to indicate panel is missing
- Table 2: can’t you compact the 3 tables in one?
Author Response
Reviewer 4 – Major Comments:
1) GROUP DEFINITION: Subgroups were formed for the comparisons using genetic characteristics (which ones?) and on the basis of survival (material and methods, line 478-480). Why the same groups were analyzed for EFS (results, line 93-97) as reported in the results? EFS is a value used for group definition or a variable to evaluate?
> We used the subgroups defined by the owners of the TCGA and TARGET datasets. Patients were then assigned into low and high EFS subgroups for DGE analysis and survival analysis was used to test EFS association with GOI expression for any GOI were significantly differentially expressed. However no significant results were found for any GOI within these when performing survival analysis as detailed in the results section.
2) COMPARISON: the authors asserted that gene expression differences were examined in a cohort of adult AML in comparison to pediatric/adolescent AML.
However the comparison was carried out among different subgroups in adult database and among different subgroups in pediatric/adolescent database, but a direct comparison between adult pediatric/adolescent was not described.
> Due to differences between datasets – patient characteristics, clinical info such as treatment, definitions for EFS, etc. a direct comparison was not possible (for example, from DGE analysis onwards, we only looked at similarities at the end by doing Venn diagrams).
Minor Comments:
1) The introduction section is too long. It should be shortened and help the common reader to better understand this difficult analysis.
> The Introduction is 50 lines long and edited to help the ‘common’ reader understand the analysis.
2) Both in paragraphs 2.1.1 and 2.1.2, the authors listed the prognostic subgroups in which the analysis did not allow to continue a focused analysis on differential expressed GOI. Nevertheless the 3 subgroups identified for further analysis were not explicitly specified. A specification could help in reading results.
> The subgroups that were identified for further analysis are now stated. Lines 113 and 114; and 170-172.
3) Figure 1: the letter “B” and (i) to indicate panel is missing
> Now visible.
4) Table 2: can’t you compact the 3 tables in one?
> Table 3 (line 283) is now compacted into 1 Table.